# Use of In Vitro Dynamic Colon Model (DCM) to Inform a Physiologically Based Biopharmaceutic Model (PBBM) to Predict the In Vivo Performance of a Modified-Release Formulation of Theophylline

**DOI:** 10.3390/pharmaceutics15030882

**Published:** 2023-03-09

**Authors:** Konstantinos Stamatopoulos, Connor O’Farrell, Mark J. H. Simmons, Hannah K. Batchelor, Nena Mistry

**Affiliations:** 1Biopharmaceutics, DPD, MDS, GSK, David Jack Centre, Park Road, Ware SG12 0DP, UK; 2School of Chemical Engineering, University of Birmingham, Edgbaston, Birmingham B15 2TT, UK; 3Strathclyde Institute of Pharmacy and Biomedical Sciences, University of Strathclyde, 161 Cathedral Street, Glasgow G4 0RE, UK

**Keywords:** PBBM, PBPK, Dynamic Colon Model (DCM), SimCyp^®^, in vitro model, dissolution testing, pharmacokinetics, in silico modelling, theophylline, modified release (MR), USP II

## Abstract

A physiologically based biopharmaceutic model (PBBM) of a modified-release formulation of theophylline (Uniphyllin Continus^®^ 200 mg tablet) was developed and implemented to predict the pharmacokinetic (PK) data of healthy male volunteers by integrating dissolution profiles measured in a biorelevant in vitro model: the Dynamic Colon Model (DCM). The superiority of the DCM over the United States Pharmacopeia (USP) Apparatus II (USP II) was demonstrated by the superior predictions for the 200 mg tablet (average absolute fold error (AAFE): 1.1–1.3 (DCM) vs. 1.3–1.5 (USP II). The best predictions were obtained using the three motility patterns (antegrade and retrograde propagating waves, baseline) in the DCM, which produced similar PK profiles. However, extensive erosion of the tablet occurred at all agitation speeds used in USP II (25, 50 and 100 rpm), resulting in an increased drug release rate in vitro and overpredicted PK data. The PK data of the Uniphyllin Continus^®^ 400 mg tablet could not be predicted with the same accuracy using dissolution profiles from the DCM, which might be explained by differences in upper gastrointestinal (GI) tract residence times between the 200 and 400 mg tablets. Thus, it is recommended that the DCM be used for dosage forms in which the main release phenomena take place in the distal GI tract. However, the DCM again showed a better performance based on the overall AAFE compared to the USP II. Regional dissolution profiles within the DCM cannot currently be integrated into Simcyp^®^, which might limit the predictivity of the DCM. Thus, further compartmentalization of the colon within PBBM platforms is required to account for observed intra-regional differences in drug distribution.

## 1. Introduction

Predicting the in vivo behaviour of modified-release (MR) formulations is challenging [1]. This is because the dissolution of drug from MR formulations is not solely dependent on the properties of the drug, but on those of the formulation, specifically the effect of the polymeric or other type of excipients used to achieve the desired release rate.

It has been shown that the hydrodynamics and mechanical forces exerted from gut wall motion in the gastrointestinal (GI) tract can significantly affect drug release from oral MR formulations in vivo [2]. This is particularly the case for those MR formulations where the triggering mechanism to release the drug is based on the erosion of the polymeric gel formed upon hydration of a hydrophilic matrix.

Unlike the fixed hydrodynamic conditions in a compendial in vitro dissolution apparatus, dosage forms will pass through a dynamic and diverse hydrodynamic environment along the GI tract. In particular, the transit of drug product is characterized by relatively long static phases, interrupted by phases of elevated motility or short spikes [3] at variable velocities and pressures depending on the region of the GI tract [4,5,6].

MR dosage forms will be exposed to this diverse environment during passage through the GI tract [1]. In particular, dosage forms that aim to release the drug in the distal/lower part of the GI tract will be exposed to a unique environment compared to the upper GI tract, in terms of fluid volumes [7], the presence of gut microbiota [8], complex motility patterns [9], osmolality [10], viscous contents [11], a thick mucus layer [12] and pH [13,14].

Although confidence in the simulation of the dissolution of drug particles from immediate-release (IR) dosage forms is reasonably high, the lack of well-established mechanistic models limits the application of bottom-up physiologically based biopharmaceutic modelling (PBBM) of MR formulations. This is because a mechanistic understanding of the complex interplay of polymers, drugs, gastrointestinal (GI) components (e.g., buffer species and bile salts) and hydrodynamics (e.g., motility and shear forces) as well as food effects (e.g., viscosity, fat, protein and fibre) is required, which is not an easy task. Thus, the current approaches to simulating the in vivo dissolution of drug particles from MR formulations are (a) to use the in vitro compendial dissolution profile as a release profile and allow mechanistic models (e.g., Advanced Dissolution Absorption Metabolism (ADAM) in Simcyp^®^ or Advanced Compartmental Absorption and Transit (ACAT^TM^) in Gastroplus^®^) to handle the dissolution of the released drug particles [15,16]; (b) to directly import the dissolution profile into the PBBM (although this is not recommended as it removes inter-subject variability from the simulations); and (c) to fit the in vitro dissolution profile, mainly derived from USP apparatuses, using a Weibull function [17,18]. Thus, the most important input to the PBBMs for MR formulations is still the in vitro dissolution profile.

In acknowledgement of the importance of accounting for in vivo hydrodynamic/mechanical conditions in understanding the in vivo behaviour of MR formulations, several dynamic in vitro tools have been developed mimicking different parts of the human GI tract, such as TIM1 [19,20], the rotating beaker apparatus [21], the dynamic gastric model [22] and the dissolution stress test device [23,24]. These advanced dynamic in vitro tools have been key facilitators in the understanding of irregular absorption profiles observed from MR formulations in vivo [23], identification of suitable MR formulations [25] and in achieving in vivo–in vitro correlations (IVIVCs) [20,26].

However, an in vitro tool aiming to mimic the colonic region of the human GI tract must reproduce the segmented (haustra) colon wall, which impacts propulsion of the contents [27], the motility patterns, distribution of the dissolved drug [28], fluid volumes and viscosity as well as the presence of microbiota. TIM-2 is a computer-controlled advanced multicompartmental in vitro tool [29] that combines gut microbiota and simple peristaltic motion [30] using a straight flexible tube. So far, this model has been mainly used to explore food effects on the microbiota [31].

The Dynamic Colon Model (DCM), developed by Stamatopoulos et al. [32], was designed to mimic the environment in the proximal colon [30], and its biorelevancy has been extensively characterized and validated in terms of architecture, pressure amplitudes [33], motility patterns [32] and hydrodynamics [34]. However, the clinical relevance (or biopredictivity) of in vitro observations made using the model is still lacking.

Previous studies have demonstrated the benefits of integrating dynamic in vitro dissolution tools (e.g., TIM-1) into PBBMs to support formulation selection and predict/understand food effects observed in clinical studies [35]. However, the integration of advanced dynamic in vitro tools in in silico models is limited. Hence, PBBMs still rely on compendial in vitro dissolution methods to derive empirical parameters to inform simulations for MR formulations.

The aim of this work was to incorporate the dissolution profiles of an MR formulation measured in the DCM into a PBBM to predict clinical PK data. Since there is no absorptive component in the DCM, theophylline was selected as the model drug to prevent any potential solubility/permeability-limited processes, which could have affected the dissolution rate in the in vitro model. The Uniphyllin Continus^®^ tablet (Napp Pharmaceuticals) containing theophylline was selected to explore the impact of colonic hydrodynamics on erosion and drug release. The in vitro dissolution profiles were imported into the PBBM to predict clinical data available in the literature. Furthermore, dissolution profiles of Uniphyllin Continus^®^ derived from the United States Pharmacopeia (USP) Apparatus II (USP II) at different agitation speeds were also imported in the PBBM, and the predictions were compared with those obtained using the DCM.

## 2. Materials and Methods

### 2.1. Dissolution Data

O’Farrell et al. [36] generated dissolution profiles of the Uniphyllin Continus^®^ tablet (200 mg) obtained from the USP II filled with 900 mL of pH 7.4 phosphate buffer at 25, 50 and 100 rpm and from the DCM filled with 900 mL of 0.25% (*w*/*v*) NaCMC solution buffered to pH 7.4, under six different motility patterns [36]. The authors showed that the formulation is sensitive to agitation speed and hence it was selected as a case study to assess the clinical relevance of the in vitro hydrodynamics of the DCM.

Following the same protocol, dissolution experiments were also conducted for Uniphyllin Continus^®^ tablets (400 mg) using USP II (900 mL) and the DCM. However, just two different motility patterns were used in the DCM: baseline and a retrograde propagating wave. To ensure solubility challenges were avoided, the sample volume was doubled from 1 mL per sample point per time point to 2 mL. Media replenishment was modified accordingly to account for this change. Theophylline concentration was measured using UV-Visible spectrophotometry (Biochrom Libra S12) at 270 nm using a quartz cuvette with a 10 mm optical path at 22 °C. Absorbance at 270 nm was compared against a calibration curve that was linear between 2 and 20 μg mL^−1^ (correlation coefficient of 0.999) obtained through testing a serial dilution of standard solutions (*n* = 4).

### 2.2. Development of PBBM

The PBBM was built using the Simcyp^®^ Simulator v20 (Certara Limited, Simcyp division, Sheffield, UK). All simulations were carried out using the healthy volunteer population. The PK parameters of theophylline (Appendix A) were derived from the non-pregnant populations in Abduljalil et al. [37].

#### 2.2.1. Modelling Human Colonic Absorption

Staib et al. [38] assessed the absorption of theophylline solution at different regions of the colon (ascending, descending and sigmoid colon) in 3 male healthy volunteers using a remote-controlled capsule. The location of the capsule was confirmed using X-ray.

To simulate intra-luminal dosing of theophylline solution directly in the proximal colon of healthy human volunteers [38], a per os (PO) simulation was run with the transit times of the ADAM model compartments prior to the colon being set to 0.001 h. This change allowed for “instantaneous” dosing of theophylline to the colon. This setting was used only to simulate intracolonic dosing. The mechanistic permeability (MechPeff) model was used to provide regional permeability and absorption scalar, whilst transit times in the colon compartment were optimized to capture the observed PK data published by Staib et al. [38] following intracolonic administration of theophylline solution.

#### 2.2.2. Integrating the Dissolution Data

There are two ways for the ADAM model to handle in vitro dissolution/release profiles of MR formulations, either as a dissolution profile or as a release profile [15]. In the first case (dissolution profile), the model converts the mass of the API in the MR formulation to dissolved drug according to dissolution rate over time. In the second case (release profile), the model makes the mass of the API available for dissolution based on the release profile input. In this case, the dissolution of the released particles at each time point of the simulation is handled by the Diffusion Layer Model (DLM), which accounts for the solubility and particle size of the active pharmaceutical ingredient (API).

In this work, the dissolution profiles of the Uniphyllin Continus^®^ tablet (200 mg and 400 mg) obtained from the USP II and DCM were imported into the ADAM model as discrete release profiles using linear regression between the data points (this option was available in the Simcyp^®^ simulator without any further modification by the authors).

#### 2.2.3. Prediction Accuracy

The AAFE (average absolute fold error) was calculated as described by Shimizu et al. [39] to statistically compare the observed and simulated pharmacokinetic parameters as well as the observed/predicted ratio. The ratio of observed to predicted values for PK parameters was used to assess the performance of the PBBM.

## 3. Results

### 3.1. In Vitro Dissolution Profiles of Uniphyllin Continus^®^ Tablet (400 mg) Measured in the DCM and USP II

Figure 1 shows the dissolution profiles of the Uniphyllin Continus^®^ tablet (400 mg) obtained from the DCM and USP II under different motility patterns and agitation speeds, respectively. As expected, the rate and extent of release were highest in the USP II at 100 rpm. Release slowed after 16 h approaching a plateau that reached 98.1% release at 24 h. Under all other conditions, the release steadily increased over the period of 24 h in a linear fashion with no sign of reaching a plateau. The lowest release was exhibited under the DCM at baseline, whilst the retrograde pattern was almost indistinguishable from the profile measured in the USP II at 25 rpm. Lower release was observed in comparison to the 200 mg tablet (see Figure 1). Locally to the tablet body, higher concentrations of drug were present, which may have hindered diffusion of theophylline through the gel matrix into the surrounding media. Additionally, the 400 mg tablets are larger (16.0 × 7.0 × 5.1 mm, 550 ± 10 mg, *n* = 3) than the 200 mg tablets (11.7 × 5.5 × 4.1 mm, 265 ± 10 mg, *n* = 3). This ultimately presents a longer diffusion pathway for theophylline, as the distance for water to penetrate and the time taken to form a gel layer that extends to the core of the tablet would be greater. Furthermore, more polymer material needs to be eroded in order to achieve complete release under the same hydrodynamic conditions.

### 3.2. Modelling Human Colonic Absorption

Initially, Abduljalil et al.’s [37] theophylline model (which is also available in Simcyp’s compounds library) was used, and its predictivity was assessed against Staib et al.’s [38] in vivo data of the theophylline solution administered directly to the colon. The only modification was to block the absorption of theophylline in the stomach and small intestine (SI) (by setting the absorption rate scalar (ARS) to 0.001, as well as setting the transit times in these two regions to 0.001 h), as explained in Section 2.2.1. The model underpredicted the absorption rate, the maximum plasma concentration of theophylline (C_Max_) and the rate of reduction in the theophylline plasma concentration of the observed data (Figure 2a), and hence further investigation was conducted.

Parameter sensitivity analysis (PSA) was performed to investigate how transit times and colonic absorption affected the predictions. Instead of changing the human intestinal steady-state effective permeability in the colon, the P_eff,man_ value, as predicted by MechPeff, the ARS was used in the PSA. The reason for using the MechPeff model to predict the regional P_eff,man_ was based on the anatomical differences in the GI tract and to account for inter-subject variability in gut physiology. Then, the ARS can be applied to scale up/down the total absorption in each intestinal compartment. PSA showed that the ARS in the colon mainly affects the exposure of theophylline, whereas the transit times did not. Although the absorption phase of the plasma concentration (C_p_) profile was captured by the model, the model overpredicted the level of exposure of theophylline (Figure 2b). It seems that this overprediction was mainly related to the drug distribution phase in the plasma profile, which was not captured by the model.

The data of Staib et al. [38] were not included in the clinical studies of Abduljalil et al. [37] to develop and verify their PBPK model. However, their model showed good predictivity across an array of clinical studies involving oral administration of theophylline solution. Thus, the oral administration of the theophylline solution arm of the study by Staib et al. [38] was applied as a control study in this work when optimizing for colonic absorption, guided by the intracolonic dosing arm of the study. In this case, the ARS of the stomach and SI was reset to the default value of one, and the transit times back to their default values. Still, the model overpredicted the observed data, although the observed data fell between the 5th and 95th percentile of the prediction (Figure 2c).

Considering the small scale of the study (*n* = 3) by Staib et al. [38] compared to the validated model of theophylline in the Simcyp library (*n* = 32, 3 clinical trials), there were two options to refine the predictive power of the simulation. The first one was to develop a bespoke fit-for-purpose model using the Staib data; however, it would have been challenging to extrapolate this model to other clinical studies and predict the exposure of theophylline following administration of MR formulations. The second option was to use the PK parameters of theophylline and the ARS and transit times for the stomach and SI based on the model developed by Abduljalil et al. [37], and optimize the ARS in the colon so that the predicted AUC of theophylline exposure after direct administration in the ascending colon (AUC_AC_) fell within the in vivo-observed range of 79–87% of the predicted AUC in the control study (AUC_C_) [38]. Clear et al. [40] also showed that the AUC of an IR tablet of theophylline delivered directly to the colon using an Intelisite^®^ capsule was ~85% of the AUC obtained after oral administration of the same IR tablet. However, this study also had a small sample size (*n* = 3).

Hence, the second option was followed, assuming that the model of theophylline developed and validated by Simcyp^®^ is appropriate for the upper GI tract and that the PK parameters (kinetic disposition, elimination half-lives, etc.) do not change based on the site of absorption (this is also supported by Staib et al. [38]). Thus, the ARS in the colon was set to 1.75 in the model (and this value was also used in the following simulations), which achieved a predicted mean AUC_AC_/AUC_c_ ratio of 0.85, i.e., within the range of the AUC_AC_/AUC_c_ ratio observed in vivo [38,40].

### 3.3. Simulation of Human Pharmacokinetic Data

The in vitro dissolution profiles of the Uniphyllin Continus^®^ tablet (200 mg) derived from the USPII and DCM were imported separately into the ADAM model as release profiles, and the predictivity of the PBBM was assessed against the clinical data.

Figure 3 shows the predictions of theophylline Cp–time profiles after administration of the Uniphyllin Continus^®^ 200 mg tablet bis in die (b.i.d) for 5 days, with a single dose on day 6 [41] using the USP II and DCM dissolution profiles as published by O’Farrell et al. [36].

Using three motility patterns—baseline, antegrade and retrograde—from the DCM in vitro data, the model could predict the in vivo data (Figure 3b’). However, the PBBM overpredicted the in vivo data when the motility pattern that mimics the stimulated conditions in a human colon after administration of maltose solution was incorporated into the ADAM model.

No models that integrated the USP II in vitro dissolution profiles were able to predict the observed data with good accuracy as all overpredicted the in vivo data.

Figure 4 shows the predicted regional fraction absorbed when guided by dissolution profiles from the USP II and DCM. The colon was the region in which most of the drug was absorbed. Overestimating the release rate in the SI, and hence the fraction of the drug absorbed (f_a_), led to overpredicting the in vivo data. This can be seen in Figure 4, which indicates the fraction absorbed in the compartments of the SI was higher when the dissolution profiles from the USP II were used, compared to the fraction absorbed using the dissolution profiles from the DCM. However, overestimation of the release rate within colon can also lead to overpredictions. The fraction absorbed in the colon was 0.6 when the DCM–maltose motility pattern was incorporated into the model, which led to overpredicting the PK data of theophylline compared to the fraction absorbed values of 0.35 and 0.41 for the baseline and antegrade motility pattern, respectively.

The overall simulated f_a_ was 0.65, 0.76 and 0.87 according to the 25, 50 and 100 rpm USP II dissolution experiments, respectively, whereas for the DCM, the overall simulated fraction absorbed was 0.50, 0.52, 0.59 and 0.78 for the baseline, antegrade, retrograde and maltose motility patterns, respectively. Clear et al. [40] showed that the losses of theophylline (i.e., amount of drug recovered from the Intelisite capsule post-defaecation), in most participants, were in the range of 34–54%. These losses can be captured from 25 rpm as well as from baseline, antegrade and retrograde release profiles. Although it is important to capture the overall f_a_ for MR formulations, it is also important to accurately capture the regional f_a_.

Figure 5 presents the predicted plasma concentration profiles following oral administration of the Uniphyllin Continus^®^ tablet (400 mg) (single dose) when the dissolution profiles from the USP II (100 rpm) and DCM (retrograde) were included. Both in vitro dissolution profiles resulted in underprediction of the data points at t ≤ 12 h but overprediction of the plasma concentration of theophylline at t = 24 h due to slower release and therefore clearance from the systemic circulation. Most data points fell within the 5th to 95th percentile of the mean prediction when using the USP II at 100 rpm, suggesting that more intense hydrodynamics are required to predict the in vivo performance of this tablet, assuming solubility effects are insignificant.

Figure 6 presents the predictions for the 400 mg Uniphyllin Continus^®^ tablet (twice daily (b.i.d) for 4 days) using the dissolution profiles from the DCM and USP II (Figure 1). Unlike the 200 mg tablet predictions (Figure 3), the PBBM underpredicted the observed data of the 400 mg tablet (b.i.d) (although the data fell between the 5th and 95th percentile) when the in vitro dissolution profiles derived from the DCM baseline and retrograde motility patterns were integrated into the model. Based on the USP II dissolution profiles, aggressive erosion should be applied to improve the predictions, and this can be seen from the prediction performance of the model when the 100 rpm dissolution profile was used (Figure 6). Although the colon remained as the region in which most of the drug was absorbed, more drug needed to be released and absorbed in the upper GI tract in order to capture the observed data. This can be seen from the regional f_a_ profiles of the DCM baseline and USP II 100 rpm (Figure 7). In particular, 30% of the drug was absorbed in the small intestine with the USP II compared to 17% with the DCM. This requires a significant amount of the drug to be released within the small intestine. Approximately 36% of the drug was released within 6 h with the USP II at 100 rpm but only 15% with the DCM baseline dissolution profile (Figure 1). Thus, a preliminary upper GI dissolution step may be required in vitro prior to introducing the tablet to the DCM for this formulation.

Table 1 summarizes the performance of the PBBM using the dissolution profiles from the DCM and USP II for 200 and 400 mg Uniphyllin Continus^®^ tablets. However, not all dissolution profiles allowed the PBBM to meet the acceptance criteria (0.5- to 2-fold predicted/observed [42]) for all PK parameters. The 200 mg tablet (multiple ascending dose; MAD) was best predicted by the baseline, antegrade and retrograde DCM motility patterns (see Pred/Obs and AAFE value comparison in Table 1). The DCM did not predict the PK parameters for the single dose (SD) and for the multiple ascending dose (MAD) of the 400 mg tablet with the same accuracy. The model had a better performance for the 400 mg tablet (MAD) with the in vitro dissolution profile derived from USP II with an agitation speed of 100 rpm. However, the Cp–time profile was a poor fit for both approaches to the 400 mg tablet after single-dose administration (Figure 5).

## 4. Discussion

In this work, the in vitro dissolution profiles derived from a biorelevant Dynamic Colon Model (DCM) and from the USP II were incorporated into a PBBM, and the predictions were assessed against previously published in vivo data for Uniphyllin Continus^®^ tablets.

Since there is no absorptive component in the DCM, theophylline was selected as the model drug to avoid any potential solubility/permeability-limited processes, which could have affected the dissolution rate in the in vitro model.

The most challenging part of this study was to model the colonic absorption of theophylline. This is because Simcyp v20 provides a single compartment of the colon, which averages the absorptive surface area, fluid volumes and transit times. Although the segregated transit time model provides the option of applying either the residence times of the whole colon or those of the ascending region, these options are not linked to the corresponding anatomical differences in terms of diameter and length. Thus, the only option to overcome this limitation was to adjust the absorption in this region by scaling it up with the ARS and to make sure that the AUC_AC_/AUC_c_ ratio fell within the observed range in vivo.

The DCM proved to be biopredictive for the Uniphyllin Continus^®^ tablet, whereas the USP II was not biopredictive at 25, 50 or 100 rpm. However, the assumption here was that the hydrodynamics in the upper GI tract are not relevant or impactful to this formulation and that most of the erosion takes place within the colon. Another assumption was that the in vitro dissolution profile in the DCM represents the overall in vivo release profile of the entire colon (although the DCM is designed to replicate the proximal colon only). This assumption may stand for most cases, as MR formulations are designed to release drugs within this region since there is more water available to dissolve drugs as well as a higher absorption surface area compared to the distal part of the human colon.

The simulations also showed that three different motility patterns (baseline, antegrade and retrograde) could explain the level of exposure of theophylline after oral administration of the Uniphyllin Continus^®^ tablet. However, O’Farrell et al. [36] showed differences in the distribution of theophylline throughout the DCM after applying different motility patterns. The average dissolution profiles resulting from the baseline, antegrade and retrograde waves were similar. However, the regional in vitro dissolution profiles of those three motility patterns showed higher accumulation of theophylline at the beginning of the DCM tube, the highest being observed in the baseline motility pattern. A more homogenous distribution of theophylline was observed with the antegrade and retrograde propagating motility patterns, as these patterns allowed dissolved drug to be transferred to the middle and distal regions of the DCM tube.

However, determining how these differences in drug distribution along the DCM might impact the in vivo performance is challenging, as commercially available PBPK platforms do not allow the user to incorporate regional in vitro dissolution profiles into the colonic compartment. The addition of such compartments in the colon is of value for MR formulations.

In an attempt to translate in vitro observations to in vivo absorption, we could note that the baseline motility may not be sufficiently erosive and causes a high accumulation of the drug within the caecum (beginning of the) ascending region, leading to an elevated concentration gradient across the gut wall, hence not significantly reducing the absorption rate. On the contrary, antegrade or retrograde propagating waves with low amplitude that are also not particularly erosive (that generate low shear rates on the tablet surface) improve the distribution of the drug along the ascending region, exposing it to a greater absorptive surface area, at the cost of diluting the drug and lowering regional concentration gradients across the gut wall. However, the impact of this regional variation due to physiological motility and geometry on overall absorption needs further investigation.

Another factor that remains to be proven is whether the baseline, antegrade and retrograde waves reflect the motility patterns in a healthy human population over an extended period, so they can be implemented as standard in vitro dissolution steps used to assess the in vivo performance of MR formulations. For this purpose, more dissolution experiments using MR formulations with different release rates and/or mechanisms should be conducted using the DCM with the aim of developing in vitro–in vivo correlations supported by physiologically based biopharmaceutic modelling.

Unlike those of the 200 mg tablet, the dissolution profiles of the 400 mg tablet derived from the DCM seemed to be less biorelevant and biopredictive of the Cp–time profiles (although the PK parameters were predicted in most cases within the acceptance criteria). In this case, a more aggressive erosion of the tablet is required to capture the in vivo data. In particular, the dissolution profile obtained with USP II at 100 rpm helped to capture the Cp–time profile of theophylline. A potential reason for this discrepancy between the two tablets might be due to differences in residence time along the GI tract. The 400 mg tablet is bigger compared to the 200 mg tablet, so it might remain in the stomach and the small intestine for a longer period. Furthermore, big tablets that do not rapidly disintegrate are emptied from the stomach mainly due to the strong, high-amplitude motility patterns of migrating motor complexes (MMCs) in fasted subjects [43]. Thus, these dosage forms may experience greater levels of erosion in vivo, resulting in the release of a greater amount of the API in the small intestine to be made available for absorption. Hence, using the DCM to describe the in vivo dissolution profile of big MR tablets might not be appropriate, and a preconditioning dissolution step (e.g., using USP II with media reflecting the gastric and small intestinal conditions) should be added to the protocol before the tablet is introduced into the DCM to simulate dissolution in the lower GI tract as a final step. This will also better reflect the clinical administration protocol in which an MR tablet is normally given orally, where it will first pass through the stomach and then the small intestine before reaching the colon.

## 5. Conclusions

A PPBM of an MR formulation of theophylline (Uniphyllin Continus^®^ tablet) was developed and successfully used to predict PK data observed in healthy male volunteers. This was possible using the in vitro dissolution profiles of theophylline derived from the biorelevant Dynamic Colon Model (DCM). The ability of the DCM to reproduce the in vivo release profile of the Uniphyllin Continus^®^ 200 mg tablet was demonstrated in this work for the first time, whilst the USPII dissolution profiles measured at 25, 50 and 100 rpm failed to describe the in vivo performance of the Uniphyllin Continus^®^ tablet.

Differences in the distribution of the drug along the DCM between different motility patterns might not be clinically relevant, as using the average in vitro dissolution profile of three different motility patterns (baseline, antegrade and retrograde) could explain the level of exposure of theophylline after the oral administration of the Uniphyllin Continus^®^ tablet. However, “aggressive” motility patterns, such as maltose, mimicking stimulated conditions in the human colon, can lead to overestimation of the erosion rate of the formulation and cause the PBBM to overpredict PK parameters.

The PK data of the Uniphyllin Continus^®^ 400 mg tablet could not be predicted with the same accuracy using the dissolution profiles from the DCM. Differences in the residence times along the GI tract between the 200 and 400 mg tablets might explain why the DCM was not able to better describe the in vivo disintegration and dissolution of the 400 mg tablet. Thus, it is recommended that the DCM be used mainly for dosage forms, for which the main erosion, disintegration and dissolution processes take place in the distal part of the GI tract.

It was demonstrated that while it is important to capture the overall fraction absorbed for MR formulations, it is also important to properly capture the regional fraction absorbed.

Whether the baseline, antegrade and retrograde all reflect the motility patterns in a healthy human population and can be used as standard in vitro dissolution values to assess the in vivo performance of MR formulations remains to be proved. For this purpose, more dissolution experiments using MR formulations with different release rates and/or erosion mechanisms should be conducted using the DCM with the aim of developing in vitro–vivo correlation supported by physiologically based biopharmaceutic modelling.

Regional dissolution profiles within the DCM cannot currently be integrated into Simcyp^®^, which might limit the predictivity of the DCM. Thus, further compartmentalization of the colon within PBBM platforms is required to account for observed intra-regional differences in drug distribution.

These findings can potentially be generalized to other highly soluble small molecules in a similar dosage form and/or with a similar release mechanism (i.e., matrix erosion) but are hard to extrapolate beyond this area.

The advantages of the DCM model with respect to its better ability to mimic human colon physiology have been extensively discussed in previous works. In this study, its ability to improve the predictive capability of the informed PBBM was also demonstrated. However, this is part of ongoing research to better understand the strengths and the limitations of the DCM tool with respect to other existing models (e.g., TIM-2). This task was beyond the scope of this study as the main focus here was to determine the clinical relevance of in vitro observations in the DCM.

Furthermore, the clinical data used in this study were derived from healthy men, and it would be important to assess in future work if the dissolution profiles of theophylline MR formulations obtained from the DCM can reflect a wider range of populations based on gender and disease state.

## Figures and Tables

**Figure 1 pharmaceutics-15-00882-f001:**
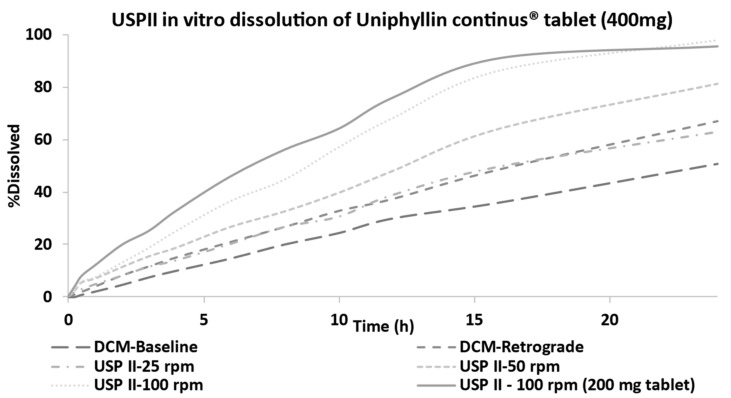
Average in vitro dissolution profiles of the Uniphyllin Continus^®^ tablet (400 mg) in the USP II (25, 50 and 100 rpm) and DCM (baseline and retrograde motility patterns).

**Figure 2 pharmaceutics-15-00882-f002:**
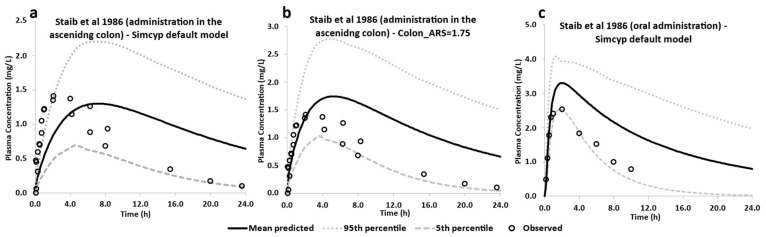
Predictions of theophylline plasma concentration profiles following (**a**) intracolonic administration of theophylline solution using the default Simcyp model (Abduljalil et al. [37]) (**b**), intracolonic dosing with optimized absorption rate scalar (ARS) in the colon compartment and (**c**) oral administration of theophylline solution (control). Clinical data were derived from Staib et al. [38], in which theophylline solution was administered to different regions of the GI tract in 3 male healthy volunteers.

**Figure 3 pharmaceutics-15-00882-f003:**
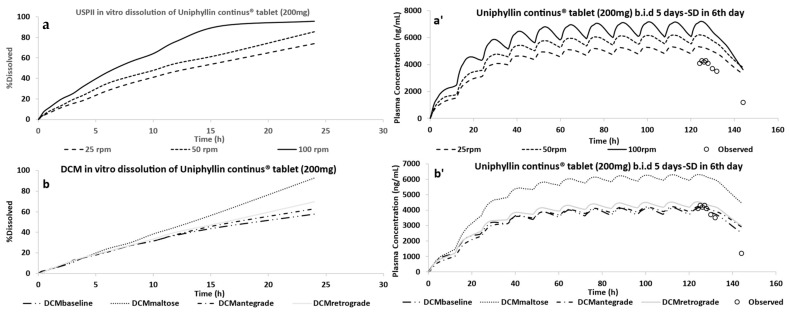
Average in vitro dissolution profiles of the Uniphyllin Continus^®^ tablet (200 mg) in the USP II (**a**) and DCM (**b**) and the corresponding predictions (**a’**,**b’**) of the plasma concentration of theophylline after administration of Uniphyllin Continus^®^ tablet (200 mg) twice daily (b.i.d) for 5 days, with a single dose (SD) on day 6 [41]. Dissolution testing applied agitation speeds of 25, 50 and 100 rpm were used in the USP II and four motility patterns were used in the DCM, as detailed in O’Farrell et al. [36].

**Figure 4 pharmaceutics-15-00882-f004:**
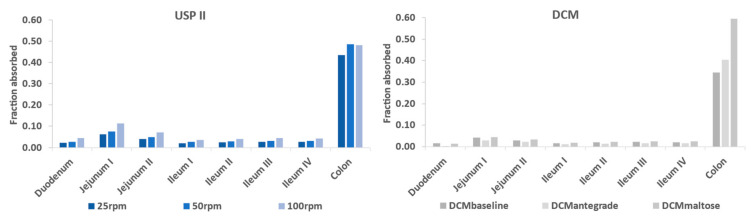
Simulated fraction absorbed from regional GI segments after incorporation of the in vitro dissolution profiles of the Uniphyllin Continus^®^ tablet (200 mg) into the Advanced Dissolution Absorption Metabolism (ADAM) model. Dissolution profiles were obtained using agitation speeds of 25, 50 and 100 rpm in the USP II and four motility patterns in the DCM, as detailed in O’Farrell et al. [36].

**Figure 5 pharmaceutics-15-00882-f005:**
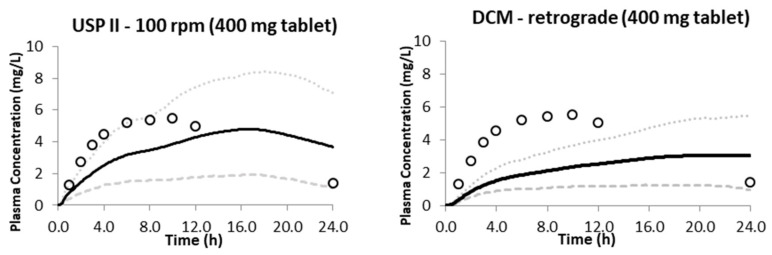
Predicted plasma concentration following oral administration of the Uniphyllin Continus tablet^®^ (400 mg) single dose (SD) with the dissolution profiles imported from the USP II at 100 rpm and the DCM with the baseline motility pattern [36]. Observed data (open circles); predicted mean (solid); 5th percentile (dashed line); 95% percentile (dotted line).

**Figure 6 pharmaceutics-15-00882-f006:**
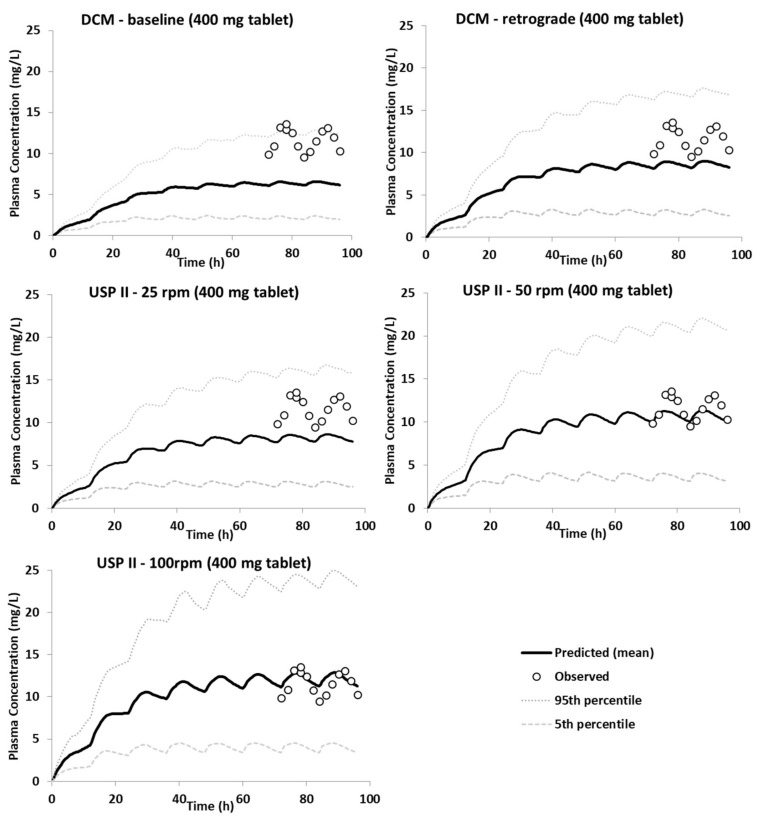
Predictions of the plasma concentration of theophylline after administration of a Uniphyllin Continus^®^ tablet (400 mg) b.i.d for 4 days using dissolution profiles obtained using agitation speeds of 25, 50 and 100 rpm in the USP II and four motility patterns in the DCM, as detailed in O’Farrell et al. [36].

**Figure 7 pharmaceutics-15-00882-f007:**
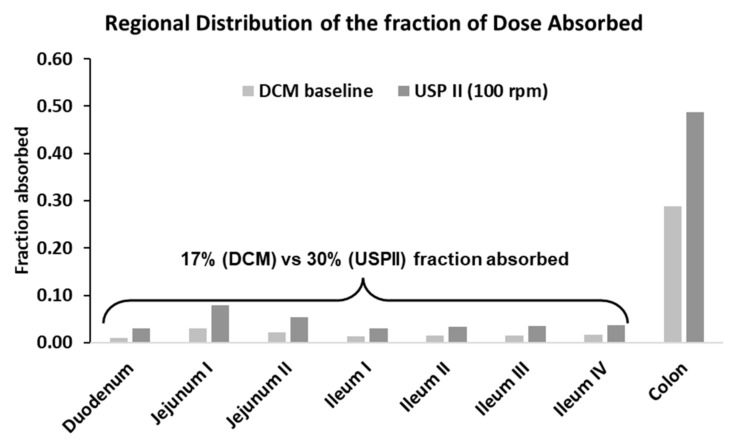
Simulated fraction absorbed from GI regions after incorporation of the in vitro dissolution profiles of the Uniphyllin Continus^®^ tablet (400 mg), derived from USP II and DCM, into the ADAM model.

**Table 1 pharmaceutics-15-00882-t001:** Mean simulated and observed pharmacokinetic parameters (C_Max_ (mg mL^−1^), AUC_0–24 h_ (mg mL^−1^ h^−1^) and T_max_ (h)) of theophylline following administration of Uniphyllin Continus^®^ tablets in healthy male volunteers. Numbers in parentheses represent the predicted to observed value ratio.

Dose		Observed	USPII (rpm)			DCM			
			25	50	100	Baseline	Antegrade	Retrograde	Maltose
200 mg (MAD)	$C_{max}$	4.3	5.3 (1.2)	6.4 (1.5)	7.2 (1.7)	4.2 (1.0)	4.2 (1.0)	4.6 (1.1)	6.3 (1.57)
	*AUC*	76.3	110.2 (1.4)	127.2 (1.7)	140.9 (1.8)	86.0 (1.13)	91.4 (1.2)	96.6 (1.27)	110.5 (1.5)
	$T_{max}$	5.0	4.0 (1.3)	4.4 (1.5)	4.8 (1.6)	4.7 (1.57)	4.5 (1.50)	4.0 (1.33)	4.4 (1.5)
400 mg (SD)	$C_{max}$	5.5	-	-	4.8 (0.9)	-	-	3.04 (0.5)	-
	AUC	115.0	-	-	87 (1.0)	-	-	54.8 (0.6)	-
	$T_{max}$	10	-	-	16.6 (1.7)	-	-	21.2 (2.1)	-
400 mg (MAD)	$C_{max}$	13.5	8.6 (0.6)	8.6 (0.6)	12.9 (1.0)	6.6 (0.5)	-	9.0 (0.7)	-
	AUC	277.5	638.5 (2.3)	832.0 (3.0)	950.5 (3.4)	481.8 (1.7)	-	104.4 (0.4)	-
	$T_{max}$	16.2	15.8 (1.0)	3.8 (0.2)	16.3 (1.0)	15.8 (1.0)	-	16.3 (1.0)	-
AAFE				200 mg		400 mg		Mean	
				USP II	DCM	USP II	DCM	USP II	DCM
			Cmax	1.5	1.1	1.3	1.8	1.4	1.5
			AUC	1.6	1.3	2.4	2.1	2.0	1.7
			Tmax	1.3	1.1	1.6	1.3	1.4	1.2

## Data Availability

The data that support the findings of this study and the code used for the simulations are freely available on request from the corresponding author.

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
