# Peer review of "Use of In Vitro Dynamic Colon Model (DCM) to Inform a Physiologically Based Biopharmaceutic Model (PBBM) to Predict the In Vivo Performance of a Modified-Release Formulation of Theophylline"

_pharmaceutics, 2023, doi:10.3390/pharmaceutics15030882_

Round 1

Reviewer 1 Report

The article shows the application of a dissolution model different from those traditionally recognized by USP. As a novel model, it seeks to generate information that allows the prediction of in vivo behavior of drugs released from different formulations. This is the novelty of the article, the application of DCM to generate information on in vivo behavior from in vitro data.

The work described in the article presents an excellent quality as novel research work. The background supports the problem to be dealt with very well, adequately describes the experimental design, including the use of previously published information, but with the appropriate references.

The discussion and conclusions are coherent and related to the experiments and the results obtained. No plagiarism was found in the paper since the authors have previously used the DCM for excellent quality papers before.

The descriptions shown, both in the results, discussion and conclusions seem clear to me, which can be seen both in the tables of results and the dissolution graphs and predictive pharmacokinetic profiles.

I consider the work as a series of excellent experiments with a high degree of novelty in the application of a new predictive model that addresses one of the main challenges of Biopharmacy.

I consider that the work can be accepted for publication.

Author Response

We would like to thank the reviewer for the kind words and we have revised the manuscript to improve the read out and to address all the reviewers' comments.

Reviewer 2 Report

Recommendation: Major Revision

Comments:

This manuscript developed and implemented a physiologically-based biopharmaceutic model (PBBM) for a MR Formulation of theophylline (Uniphyllin Continus® tablets) that successfully predicted the PK data observed in healthy male volunteers using in vitro theophylline dissolution curves derived from biologically relevant dynamic colon model (DCM). The ability of DCM to reproduce the in vivo release profile of Uniphyllin Continus® 200 mg tablets was demonstrated for the first time. The structure of this manuscript was clear. However, the content was not substantial, several issues should be addressed before the manuscript can be accepted. The main comments were listed as follows:

1.       The results showed that the in vivo performance of 200 mg theophylline sustained-release tablets could be better predicted by combining the in vitro dynamic colon model (DCM) in vitro dissolution curve with the physiology-based biopharmaceutic model (PBBM). However, there are still some questions should be solved:

1)     What was the rationale for selecting theophylline sustained-release tablets for the study, and whether the findings generalizable to other drugs.

2)     The dissolution and release of 200 mg and 400 mg of theophylline sustained-release tablets were investigated. What is the administration mode during the experiment, whether it is consistent with the clinical administration mode of theophylline, and whether it has clinical application prospects.

3)     The predictive ability of 400 mg theophylline sustained-release tablets was not significantly better than that of the United States Pharmacopeia (USP) device II (USP II), and only for theophylline sustained-release tablets whether this finding was sufficient for application.

2.       In this paper, the in vitro release data of dynamic colon model (DCM) were incorporated into PBBM to predict the in vivo release data of Uniphyllin Continus® 200 mg tablets. However, in lines 81-87 of this paper, only mentioned that the biological correlation of DCM has been widely characterized and verified in terms of structure and pressure amplitude, but the advantages of DCM model compared with other existing models have not been well reflected. Please supply the reason that DCM model was chosen requires further explanation.

3.       Theophylline a therapeutic drug for asthma. But the in vivo release data of theophylline compared in this paper are all from healthy adult male volunteers, whether this in vivo data is generally representative, and it is necessary to compare with the in vivo data of women or not.

4.       In the Section 3.1 of this paper mentions that "lower release was observed compared to 200 mg tablets." However, the dissolution curve of 200 mg theophylline sustained-release tablets is not shown in Figure 1, it is suggested to supply the relevant chart data of 200 mg preparation.

5.       In this paper, the lack of corresponding horizontal and vertical axis titles and charts in charts 2, 3, and 4 is not conducive to readers' reading and understanding. It is recommended to check the charts carefully and supplement the response content to increase the readability of the article.

6.       In this paper, the lack of corresponding horizontal and vertical axes titles and charts in figures 2, 3, and 4 in the article makes it difficult for readers to shorten the professional terms appearing for the first time, such as IR in line 58 and ADAM in line 149 without complete spelling, which makes it difficult for readers to understand the content of the article. It is suggested to check the full text and add it.

7.       Some English expressions in the manuscript, such as lines 47-51 in the text, have the problem of cumbersome expression and difficult to understand. It is suggested to further comb and improve to facilitate readers to read and understand.

8.       The reference format in the article lacks the corresponding page number. It is suggested that the author check and correct the reference format in the article by comparing with the reference format requirements of the journal.

Author Response

We would like to thank the reviewer for the valuable feedback, comments and suggestions.

Below is a point-by-point response:

Reviewer:

1)     What was the rationale for selecting theophylline sustained-release tablets for the study, and whether the findings generalizable to other drugs

Response:

We have explained the reason very clearly in the introduction (see lines 101-110).

However, we added more explanation in the discussion section (see lines 387-389) as well as extra text discussing the potential generalization to other drugs in conclusion section (see lines 492-494)

Reviewer:

2)     The dissolution and release of 200 mg and 400 mg of theophylline sustained-release tablets were investigated. What is the administration mode during the experiment, whether it is consistent with the clinical administration mode of theophylline, and whether it has clinical application prospects.

Response:

We have discussed this in the manuscript (see lines 401-406). However, we discussed it again in lines 453-458 where we added extra texted to make it clear.

Reviewer:

3)     The predictive ability of 400 mg theophylline sustained-release tablets was not significantly better than that of the United States Pharmacopeia (USP) device II (USP II), and only for theophylline sustained-release tablets whether this finding was sufficient for application.

Response:

It's not very clear what the reviewer means with this comment. However, we have explain the reasons throughout the manuscript (including the abstract) why the DCM didn't perform significantly better from USP II (100 rpm) for 400 mg tablet.

Reviewer:

2.       In this paper, the in vitro release data of dynamic colon model (DCM) were incorporated into PBBM to predict the in vivo release data of Uniphyllin Continus® 200 mg tablets. However, in lines 81-87 of this paper, only mentioned that the biological correlation of DCM has been widely characterized and verified in terms of structure and pressure amplitude, but the advantages of DCM model compared with other existing models have not been well reflected. Please supply the reason that DCM model was chosen requires further explanation.

Response:

We added extra text discussing this comment (see lines 496-502)

Reviewer:

3.       Theophylline a therapeutic drug for asthma. But the in vivo release data of theophylline compared in this paper are all from healthy adult male volunteers, whether this in vivo data is generally representative, and it is necessary to compare with the in vivo data of women or not.

Response:

Please see our response in lines 503-506. In addition, we would like to mention here that the clinical data is what was available in the public domain so we cannot control that. However, as we mentioned in line 503-506 it will be an future task to further explore. But really great suggestion!

Reviewer:

4.       In the Section 3.1 of this paper mentions that "lower release was observed compared to 200 mg tablets." However, the dissolution curve of 200 mg theophylline sustained-release tablets is not shown in Figure 1, it is suggested to supply the relevant chart data of 200 mg preparation.

Response:

The dissolution profile of 200 mg tablet obtained from USP II 100 rpm was added to the figure 1 for comparison. Please see the revise manuscript.

Reviewer:

5.       In this paper, the lack of corresponding horizontal and vertical axis titles and charts in charts 2, 3, and 4 is not conducive to readers' reading and understanding. It is recommended to check the charts carefully and supplement the response content to increase the readability of the article.

Response:

Revised

Reviewer:

6.       In this paper, the lack of corresponding horizontal and vertical axes titles and charts in figures 2, 3, and 4 in the article makes it difficult for readers to shorten the professional terms appearing for the first time, such as IR in line 58 and ADAM in line 149 without complete spelling, which makes it difficult for readers to understand the content of the article. It is suggested to check the full text and add it.

Response:

Revised

Reviewer:

7.       Some English expressions in the manuscript, such as lines 47-51 in the text, have the problem of cumbersome expression and difficult to understand. It is suggested to further comb and improve to facilitate readers to read and understand.

Response:

Revised/improved. Please lines 47-51

Reviewer:

8.       The reference format in the article lacks the corresponding page number. It is suggested that the author check and correct the reference format in the article by comparing with the reference format requirements of the journal.

Response:

The reference format is based on journal’s Endnote style downloaded from journal’s website and using Endnote software. Any further formatting will come from the journal as we cannot overwrite the journal’s Endnote style file.

Reviewer 3 Report

The paper by Stamatopoulos et al. is an interesting investigation about the possibility to predict the in vivo performance of a modified-release formulation of theophylline by biopharmaceutic modelling. The topic of the paper is worthy of investigation and well fits with the scope of Pharmaceutics. The experimental set-up is well designed and performed. The results agree with authors hypothesis and the conclusions well supported by the data.

This reviewer has no main concerns about the paper, which possesses the highs scientific standards to be published in Pharmaceutics in its current form. One suggestion for improving the paper is related to the possibility to shorten the title to better catch the interests by the readers.

Author Response

We would like to thank the reviewer for the kind words and we have revised the manuscript to improve the readout and to address all the reviewers' comments.

Round 2

Reviewer 2 Report

Recommendation: Accept

Comments:

Most of the comments have been answered, but there is still a question have not been solved, the suggestion is listed as follow:

1、Although the advantages of DCM model with respects to better mimicking the human colon physiology have been extensively discussed in previous works. More details should be supply in this manuscript.
